# In Vitro Cell Interactions on PVDF Films: Effects of Surface Morphology and Polar Phase Transition

**DOI:** 10.3390/ma14185232

**Published:** 2021-09-11

**Authors:** Marco A. Alvarez-Perez, Valentina Cirillo, Maria Giovanna Pastore Carbone, Marianna Pannico, Pellegrino Musto, Vincenzo Guarino

**Affiliations:** 1TBL-DEPeI, Universidad Nacional Autonoma de Mexico (UNAM), Mexico City 04510, Mexico; marcoalv@unam.mx; 2Institute of Polymers, Composites and Biomaterials (IPCB), National Research Council of Italy, Mostra d’Oltremare Pad.20, Viale J.F. Kennedy 54, 80125 Naples, Italy; valentina.cirillo3@gmail.com; 3Institute of Chemical Engineering Sciences, Foundation for Research, and Technology Hellas (FORTH-ICEHT), Stadiou St, Platani GR-26504, 26504 Patras, Greece; mg.pastore@iceht.forth.gr; 4Institute of Polymers, Composites and Biomaterials, National Research Council of Italy, Via Campi Flegrei 32, 80078 Pozzuoli, Italy; marianna.pannico@ipcb.cnr.it (M.P.); pellegrino.musto@cnr.it (P.M.)

**Keywords:** morphology, AFM, in vitro response, hMSC, piezoelectric materials

## Abstract

In recent years, several studies have validated the use of piezoelectric materials for in situ biological stimulation, opening new interesting insights for bio-electric therapies. In this work, we investigate the morphological properties of polyvinylidene fluoride (PVDF) in the form of microstructured films after temperature-driven phase transition. The work aims to investigate the correlations between morphology at micrometric (i.e., spherulite size) and sub-micrometric (i.e., phase crystallinity) scale and in vitro cell response to validate their use as bio-functional interfaces for cellular studies. Morphological analyses (SEM, AFM) enabled evidence of the peculiar spherulite-like structure and the dependence of surface properties (i.e., intra-/interdomain roughness) upon process conditions (i.e., temperature). Meanwhile, chemical (i.e., FTIR) and thermal (i.e., DSC) analyses highlighted an influence of casting temperature and polymer solution on apolar to polar phases transition, thus affecting in vitro cell response. Accordingly, in vitro tests confirmed the relationship between micro/sub-microstructural properties and hMSC response in terms of adhesion and viability, thus suggesting a promising use of PVDF films to model, in perspective, in vitro functionalities of cells under electrical stimuli upon mechanical solicitation.

## 1. Introduction

Smart functional biomaterials include all the materials able to support in vitro cell activities over the exposure of bioactive/biocompatible signals, thanks to the integration of additive functionalities strictly related to peculiar microscopic properties (i.e., electrical, magnetic, optical) able to directly or indirectly influence in vitro cell functions [1,2]. In the last years, the fabrication of biointerfaces that do not work only as passive support for cells is dramatically increasing, due to the need to discover innovative materials able to actively instruct “*ex novo*” the formation of the native extracellular matrix by triggering/stimulating specific intracellular cues [3].

In this view, recent studies have confirmed that electro-conductive/electro-responsive materials—alone or in combination with other dielectric ones—can efficiently participate to in vitro cell activities, by corroborating the effect of morphological (i.e., pore sizes, roughness) and/or biochemical signals (i.e., peptides, adhesive factors) usually imparted to modulate the surface and bulk properties at the interface with cells [4]. In this context, electroresponsive polymers—including current responsive and voltage responsive polymers—currently represent a great promise for the fabrication of nanostructured platforms with highly tunable properties for different applications (i.e., molecular targeting [5], cell instructive interfaces [6,7], and biosensors [8]). Piezoelectric polymers such as polyvinylidene fluoride (PVDF), able to generate a voltage under mechanical stimulation/macroscopic deformation without the support of electrodes or other external sources [9], currently represent a straightforward but efficacious route to deliver electrical cues directly to cells in vitro [10]. Indeed, the peculiar semicrystalline microscopic structure of the PVDF, including five distinct crystallite polymorphs depending upon the conformation of the repeated monomer unit (−CH_2_CF_2_−)_n_, can confer some unique properties to the substrates, including piezo-, pyro-, and ferroelectric properties [11]. In particular, β-phases with all the dipoles aligned in the same direction normal to the chain axis tend to promote the largest spontaneous polarization, thus determining strong ferroelectric and piezoelectric behavior [12] that concur to particularize the in vitro response. For instance, it has been proved that piezoelectric scaffolds can enhance neurite extension, neural differentiation and neurite outgrowth, thus supporting the regeneration of in vivo peripheral nerve [13]. Furthermore, PVDF has been successfully used as bioactive coating to modify titanium surfaces for orthopedic implants. In this case, it has been demonstrated that polarization of PVDF thin layers can promote a more remarkable adhesion and proliferation of osteoblasts and osteogenic differentiation of rBMSCs compared to a nude titanium surface [14,15]. According to previous experimental evidence, recent studies have shown that PVDF-based substrates with different morphological features had different influences on cell adhesion, proliferation and differentiation due to their surface roughness properties [3].

This work aims to further investigate these correlations by proposing a simple method to fabricate PVDF films to induce the transition from apolar to polar phases by thermal treatments of PVDF solutions at different concentrations, without any support of poling processes. Herein, we will perform a systematic study of morphological and thermal properties as a function of the process conditions used, in order to identify and correlate their effect on molecular phase polarization and, ultimately, in vitro cell response.

## 2. Materials and Methods

### 2.1. Materials and Film Preparation

Poly(vinylidene fluoride) (PVDF) (average Mn ~71,000) and analytical grade N,N-dimethyl acetammide (DMAc) were acquired from Sigma Aldrich (Milan, Italy). Briefly, 2.5 mL of PVDF solution in DMAc—50 or 200 mg/mL as concentration—was cast on to a glass plate at different temperature conditions (40 °C and 80 °C), for the preparation of different films, respectively labeled as 5% and 20%, Firstly, a metal ground—i.e., aluminum circular foil with a 30 mm diameter—was included to promote an efficient heat transfer to the solution, during the casting process. Secondly, the system was placed into a water bath to assure a more uniform distribution of temperatures along the sample volume. A casting time of 6 h was selected for all the samples to assure a complete solvent evaporation.

### 2.2. Morphological Analysis

PVDF films were investigated by scanning electron microscopy equipped with a field emission source (FESEM—Quanta200, FEI, Eindovhen, The Netherlands), working at a 10 kV accelerating voltage. Specimens were treated via gold–palladium sputter coating before the analysis. Atomic force microscopy (AFM) was adopted to quantitatively investigate the morphology of the PVDF films and the effect of solution characteristics and process parameters. Imaging was performed in tapping mode, in air and at room temperature, with a Dimension Icon instrument (Bruker Corporation, Kennewick, WA, USA). An RTESPA-150 silicone probe with a nominal tip radius of 8 nm, a typical spring constant of ~5 N/m and a frequency of ~150 kHz was used for the characterization. Images were recorded at 50 × 50 µm^2^, 25 × 25 µm^2^ and 3 × 3 µm^2^ sizes with 512 × 512 lines per image, and a scan rate of 0.9 Hz. The cross-section analysis of the NanoScope Analysis software was adopted to assess the profile of the specimens in several locations. The roughness parameter root-mean-square height of the surface (Rq) was calculated using the same software and, in order to get a statistical analysis, the measurements were repeated several times on different 1 × 1 μm^2^ region of interest boxes. Results were reported as mean value ± standard deviation (SD).

### 2.3. Chemical and Physical Analyses

Thermal properties (i.e., crystallinity) of films were evaluated by differential scanning calorimetry (DSC) using a DSC Q1000 (TA Instruments, New Castle, DE, USA). An amount of approximately 10–15 mg was sealed in a nonhermetic aluminum pan under a nitrogen atmosphere. The scans were performed, under gaseous nitrogen flow, from 25 °C up to 600 °C at 10 °C/min. Neat PVDF pellets—i.e., no thermal treatments—were used as negative control (CTR) in this study. Physical transitions of PVDF were observed via Fourier transform infrared spectroscopy (FTIR). FTIR spectra were collected in the attenuated total reflection (ATR) mode using a single reflection ATR accessory (UATR from Perkin-Elmer, Newark, NJ, USA). This unit was equipped with a diamond crystal as internal reflection element (IRE) and two ZnSe crystals as transfer elements. The IRE had horizontal geometry, and an angle of incidence of 45°. The optimum wavenumber range for the present configuration was 4000–650 cm^−^^1^. The ATR accessory was equipped with a pressure transducer on the crystal tip to ensure reproducible and uniform sample-to-IRE contact. Measurements were taken by a Spectrum 100 interferometer (Perkin-Elmer), equipped with a wide-band deuterated triglycine sulfate (DTGS) detector and a Ge on KBr beam splitter. The frequency scale was internally calibrated to 0.01 cm^−^^1^ using a He–Ne reference laser. From 32 to 128 scans at a resolution of 4 cm^−^^1^ were averaged to improve the signal-to-noise ratio.

### 2.4. In Vitro Studies

In vitro biological assays were performed using a human mesenchymal stem cell line from LONZA (hMSC, PT-2501). hMSC were cultured in 75 cm^2^ cell culture flask in Eagle’s alpha minimum essential medium (α-MEM) supplemented with 10% fetal bovine serum, antibiotic solution (streptomycin 100 µg/mL and penicillin 100 U/mL, Sigma Chem. Co., Milan, Italy) and 2 mM L-glutamin. The cells were incubated at 37 °C in a humidified atmosphere with 5% CO_2_ and 95% air. Four to six hMSC passages were used for all the experimental procedures. Prior to in vitro biological assays, PVDF films were cut into a round shape, 8 mm as diameter, placed in 96 cell culture plates and sterilized by immersion in 70% of ethanol (*v*/*v*) with antibiotic solution (streptomycin 100 µg/mL and penicillin 100 U/mL), rinsed with phosphate-buffered saline (PBS) three times and air-dried. For the evaluation of cell adhesion, hMSC cells (1 × 10^4^ cells/mL) were seeded onto PVDF films and incubated for 4 and 24 h in α-MEM culture medium at 37 °C. After a prescribed time, the PVDF films were rinsed three times with phosphate buffered saline (PBS) to remove nonadherent cells. The adherent cells were fixed with 4% paraformaldehyde for 10 min and fixed cells were incubated with 0.1% Toluidine blue for 3 h was carefully washed in distilled water to remove excess of toluidine dye. The dye was extracted with 0.1% of sodium dodecyl sulfate (SDS) and the optical absorption was quantified by spectrophotometry at 600 nm with a Wallac Victor3 1420 (PerkinElmer, Boston, MA, USA). Cells in conventional polystyrene 24-well culture plates were used as a control. Cell viability of hMSC onto PVDF films was investigated in triplicate by the MTT assay at two, four, and six days of culture. This assay is based on the ability of mitochondrial dehydrogenases of living cells to oxidize a tetrazolium salt (3-[4, 5-dimethylthiazolyl-2-y]-2, 5-diphenyltetrazolium bromide), to an insoluble blue formazan product. The concentration of the blue formazan product is directly proportional to the number of metabolically active cells. hMSC seeded onto PVDF films were washed with PBS and incubated with fresh cultured medium containing 0.5 mg/mL of MTT for 4 h at 37 °C in the dark. Then, the supernatant was removed, and dimethyl sulfoxide (DMSO) was added to each well. After 60 min of slow shaking, the absorbance was quantified by spectrophotometry at 570 nm with a plate reader. The culture medium during the experimental time was changed every day with new media. Conventional polystyrene 24-well culture plates were used as a control. In order to evaluate morphology and cell spreading, hMSCs were investigated by confocal laser scanning microscopy (LSM 510, Carl Zeiss, Berlin, Germany). Briefly, hMSCs were incubated with Cell Tracker™ Green CMFDA in phenol red-free medium at 37 °C for 30 min; subsequently, hMSC were washed with PBS and incubated for 1 h in complete medium. After labeling the culture, hMSC were detached from the plate using trypsin and seeded onto PVDF films at 1 × 10^4^ cells/mL and incubated for 24 h. After the incubation time, nonadherent cells were removed by rinsing carefully three times with PBS and, then, cell–material interactions were visualized by confocal microscopy.

## 3. Results and Discussion

Organic piezoelectric biomaterials currently offer several benefits over inorganic piezoelectric materials, because they include a high biocompatibility, excellent flexibility, environmental friendliness, and a high level of processability [16]. Among them, PVDF identifies one of the most interesting candidates for use in biomedical field, due to its high dielectric constant and piezo-, pyro- and ferroelectric properties [17]. At the microscale structural level, it is characterized by different crystalline phases with different orientation and dipole packing—namely α, ε (apolar) and β, ϒ, δ (polar),—that can be appropriately balanced in order to impart unique piezoelectric properties [18]. Moreover, PVDF polymorphs can be converted by applying external physical forces—i.e., temperature, pressure, mechanical stretching—thus enabling to finely modify microstructural (i.e., crystallization) and macrostructural (i.e., roughness) properties, as widely reported in the literature [19,20].

In this work, we investigated the transition from apolar to polar phases driven by thermal treatments of PVDF solutions at different concentrations, without any support of poling processes [21].

Thermal transition is coupled to a change in surface morphology strictly dependent upon temperature conditions and polymer concentration used. This is clearly reported in Figure 1, which includes SEM images of PVDF films under different preparation conditions that confirm a relevant effect of process parameters (i.e., polymer concentration, casting temperature) on the surface morphology. In all the cases, a peculiar microstructure was recognized, including spherulite units that cover all the surface, increasing in size with the casting temperature and decreasing as solution concentration increases.

These results are confirmed by AFM analyses (Figure 2 and Figure 3). Figure 2 shows the height profile images for the different PVDF films produced in this study, and the relative profiles. It is interesting to note that the AFM analysis (i.e., interdomain measurements) confirms the spherulitic morphology highlighted by the SEM images, with the size of the spherulites increasing with the cast temperature and decreasing with solution concentration.

AFM images at higher magnification (i.e., intradomain measurements), shown in Figure 3, reveal that the spherulitic domains present a nm-rough surface, with roughness (root-mean-square height values, Rq) ranging from 8.5 ± 1.7 nm to 65.1 ± 10.9 (see Table 1). Noteworthy was that Rq increased with temperature only in the case of a less concentrated solution (5%). Indeed, in the case of 20% solution, high viscosity of the PVDF solution may tend to limit short range mass transport phenomena driven by temperature, thus inhibiting the formation of surface roughness at the nanoscale.

Further, we investigated how these morphological differences can be associated with changes in phase transition with effects on the piezoelectric properties of the film. At the first stage, differential scanning calorimetry was used for this purpose. In Figure 4, DSC thermograms showed endothermic peaks that correspond to the melting of α and β phases, falling into a temperature range from 167 °C to 172 °C, in agreement with previous studies [22].

α phases are formed by the crystallization occurring during the casting process commonly induced via solvent evaporation. In the presence of temperature, β phases characterized by all-trans planar zig-zag conformation generally occur, due to the presence of additive dipole moments triggered by thermal vibrational modes that induce locally a spontaneous polarization of crystals [22]. In this study, results highlighted that the temperature-treated films (80 °C) are characterized by higher crystallinity as confirmed by higher heat of fusion—ranging from 58.4 to 60.6 J g^−1^ as a function of the polymer concentration—with respect to the control (i.e., 52.4 J g^−1^). Similar effects were recognized also for low casting temperatures—i.e., 40 °C (Appendix A). Noteworthy was the presence of double peaks that suggested a different attitude of the PVDF solution to form crystalline phases. A unique peak was observed only in the case of high polymer concentration (20%), where the formation of multiple crystalline phases may be hindered to a reduction in polymer chains mobility occurred during the casting process [23].

In order to further investigate microscopic properties of crystalline phases, infrared spectroscopy techniques were used. It is well known that an overlapping of neighboring fluorine atoms occurs in all-trans PVDF, because the diameter of the fluorine atom (0.270 nm) is slightly larger than the space provided by an all-trans carbon chain (0.256 nm). To diminish this overlap, CF2 groups are tilted to the right and left, relative to their original conformation. This deflection of CF2 groups converts the all-trans form into TGTĞ (α form) or TTTGTTTĞ (γ form). Hence, the α-phase is more easily formed than the β- phase in normal circumstances [24]. Accordingly, ATR-FTIR analyses were performed to investigate the effect of process temperature—i.e., cast temperature—on the phase transition mechanisms (Figure 5). In particular, the IR spectrum of PVDF is very sensitive to ordering, both in terms of the overall amount of crystalline phase (crystallinity degree) and with respect to phase structure (α,γ and β polymorphic forms). According to previous works [25], there are several signatures that identify the different crystalline modifications. These are located at 410, 489, 532, 614, 763, 795, 854, 975, 1149, 1209, 1383 and 1423 cm^−1^ for the α-phase, at 445, 473 and 1275 cm^−1^ for the γ-phase and at 431, 482, 811 and 1234 cm^−1^ for the β-phase. The spectra represented in Figure 5 indicated that the PVDF sample (black trace) contains only the α-phase. Both spectra (blue and red traces, respectively) are essentially coincident: β and γ-phase are largely predominant respect to α-phase. Shoulders at 1275 and 445 cm^−1^ reveal the presence of a minor fraction of β-phase. No relevant differences in the characteristic peaks of polar phases were also detected in the case of PVDF films (5%) (Appendix A).

In order to validate the correlation between micro- (i.e., crystallinity, polar phase fraction) and macroscopic properties (roughness, spherulite size) of PVDF interfaces, in vitro tests were performed to investigate the correlations between phase transition and cell response. Figure 6A shows the in vitro response of hMSC on PVDF films in terms of adhesion, after 4 h and 24 h of cell culture. These results confirm that PVDF films cast at 80 °C, support cell adhesion better than 40 °C cast ones. Moreover, a positive effect on cell adhesion is recognized in the case of (20%) films independently upon the process temperature either at 4 and 24 h, in comparison with (5%) films. This is due to the contribution of the size effect of spherulitic interdomains (see Table 1), that confirms the strict correlation between surface roughness and cell interactions, in agreement with previous experimental evidence in the literature [26]. Accordingly, the morphology of hMSC cells was observed after 24 h in culture by fluorescence microscopy. As reported in Figure 6B, no significant differences were detected among cells, regarding cell body elongation and fibroblastic morphology. In this context, the effect of polymer concentration and casting temperature on cell viability was quantitatively evaluated via MTT assay after 2, 4, and 6 days of cell culture, by measuring the optical absorbance at 570 nm (Figure 6B). A significant increase of hMSC viability was recorded in all the PVDF films that showed higher levels of MTT conversion and spreading affinity (Figure 6C) in the case of films cast at 80 °C, independently upon the PVDF concentration used. It means that the characteristic size of spherulite-like units—at micrometric scale—obtained at higher temperatures (80 °C) tend to promote higher metabolic activities of cells on the PVDF film surface, rather than 40 °C cast ones. Furthermore, MTT data present a good statistical significance (*p* < 0.05) among different groups. Noteworthy, all the results also confirm that PVDF films, which include polar phases, showed a more efficient biological response, in terms of adhesion and hMSC viability, in agreement with previous evidences reported in literature. In perspective, this peculiar effect, due to the synergistic combination of morphological and piezoelectrical signals, could be successfully addressed to keep in vitro functionalities of tissues such as bone with intrinsic piezoelectric properties [27], in order to design electro-active models that mimic the complex microenvironment of bone under the application of in vivo-like biomechanical stimuli.

## 4. Conclusions

In this work, we have investigated morphological properties of microstructured surfaces and correlations between phase transition and in vitro cell response in order to validate the use of PVDF films as bio-functional interfaces for in vitro studies. Different spherulite-like morphologies and polar phase transitions in PVDF films have been imparted by switching casting temperature and polymer concentration. All the results revealed a more pronounced trans conformation of chains, as the polymer concentration (20%) and temperature (80 °C) increased. These peculiar morphological properties strongly influence the biological response of hMSC, affecting film biocompatibility in terms of adhesion and cell viability. In summary, micro-structured films with globular surface patterns and polar phases distribution could be really promising for the design of in vitro models to reproduce peculiar piezoelectric properties of the bone microenvironment.

## Figures and Tables

**Figure 1 materials-14-05232-f001:**
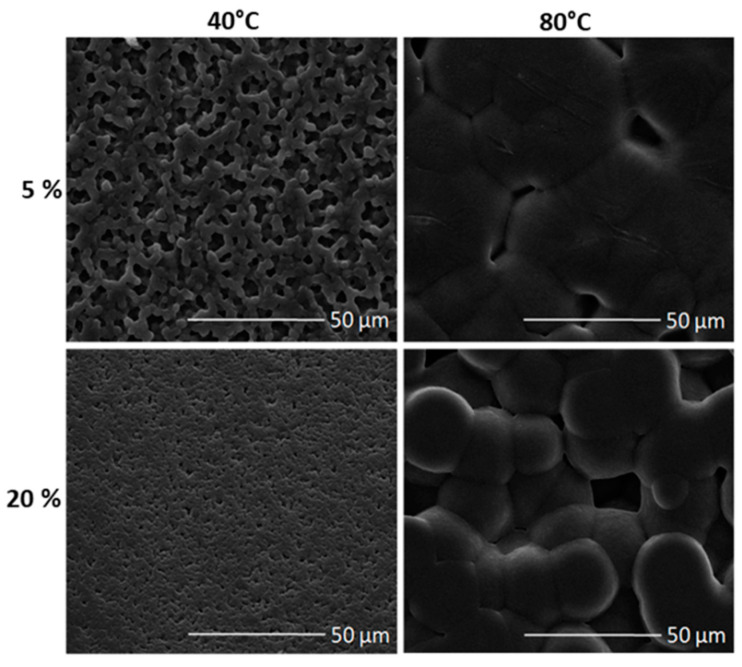
Qualitative analysis of surface morphology of PVDF films: SEM images of samples from polymer solutions with different concentration (i.e., 5 and 20%) and treated by different casting temperature (i.e., 40 and 80 °C).

**Figure 2 materials-14-05232-f002:**
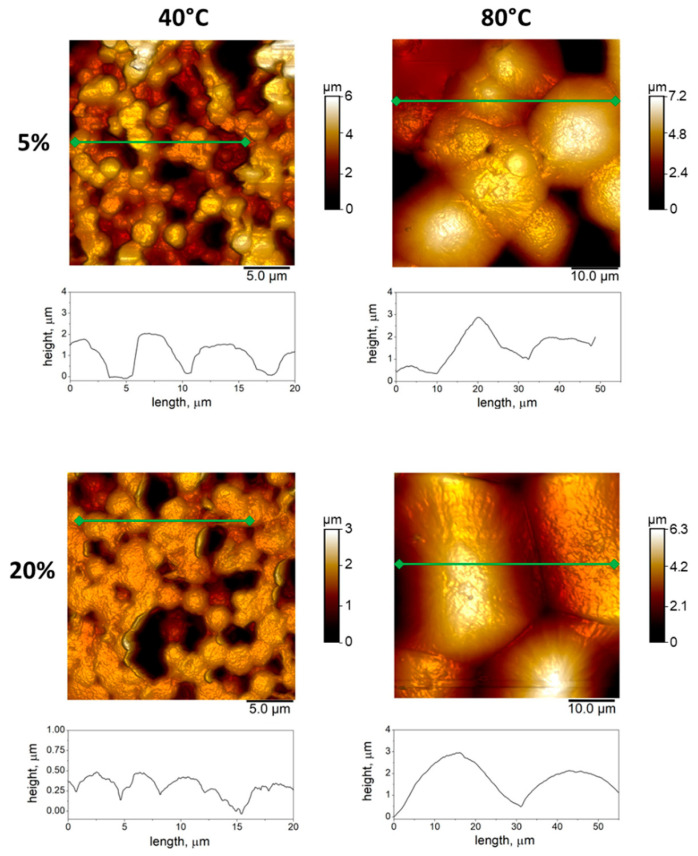
Quantitative analysis of surface morphology via AFM: images collected from samples with different polymer concentration (i.e., 5 and 20%) and treated by different casting temperature (i.e., 40 and 80 °C). Below the images, the profiles at the micrometric scale corresponding to the green cross-section lines were reported.

**Figure 3 materials-14-05232-f003:**
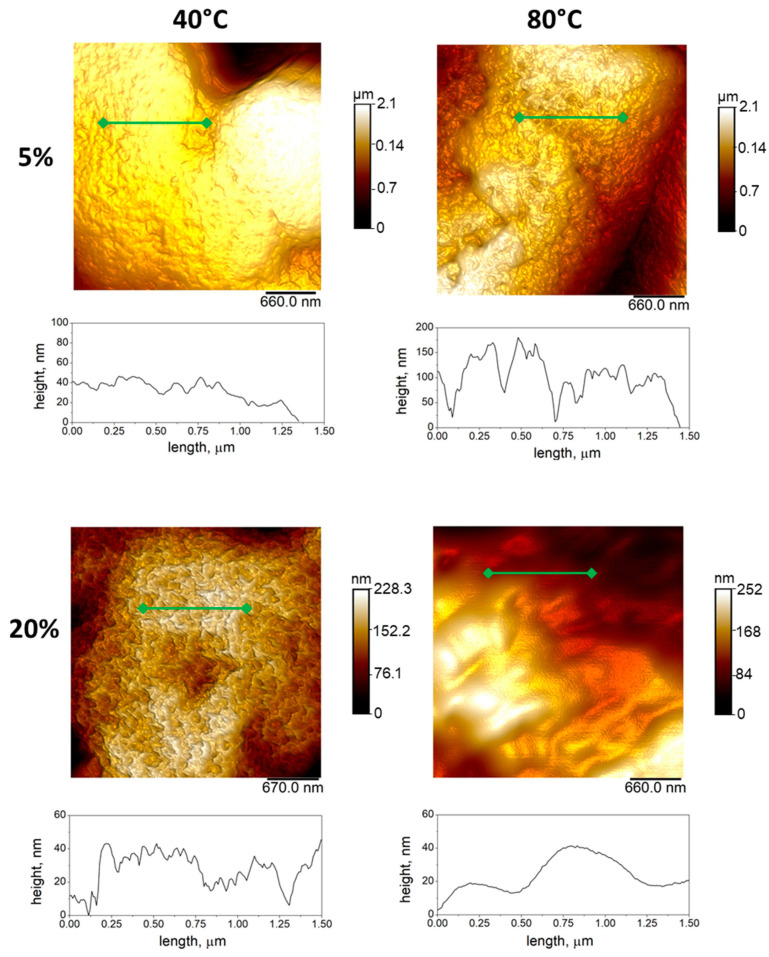
Quantitative analysis of surface morphology via AFM: images collected from samples with different polymer concentration (i.e., 5 and 20%) and different casting temperature (i.e., 40 and 80 °C). Below the images, profiles at the sub-micrometric scale corresponding to the green cross-section lines were reported.

**Figure 4 materials-14-05232-f004:**
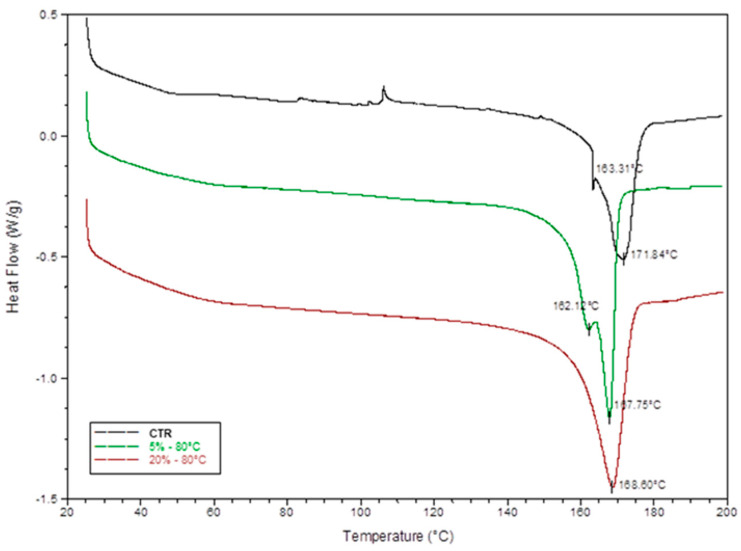
DSC thermograms of samples casted at 80 °C from PVDF solution at 5 and 20% and for pure PVDF pellets used as control.

**Figure 5 materials-14-05232-f005:**
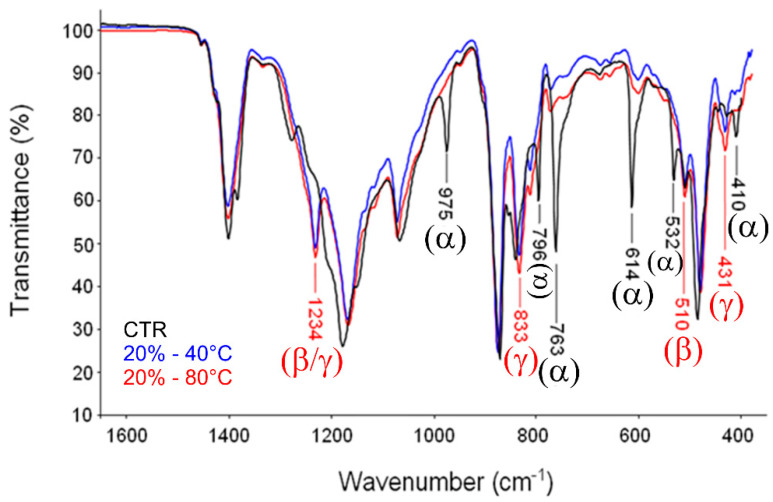
ATR spectra of PVDF films (20%) casted under different temperature conditions (40–80 °C). CTR refers to neat PVDF pellets used as control.

**Figure 6 materials-14-05232-f006:**
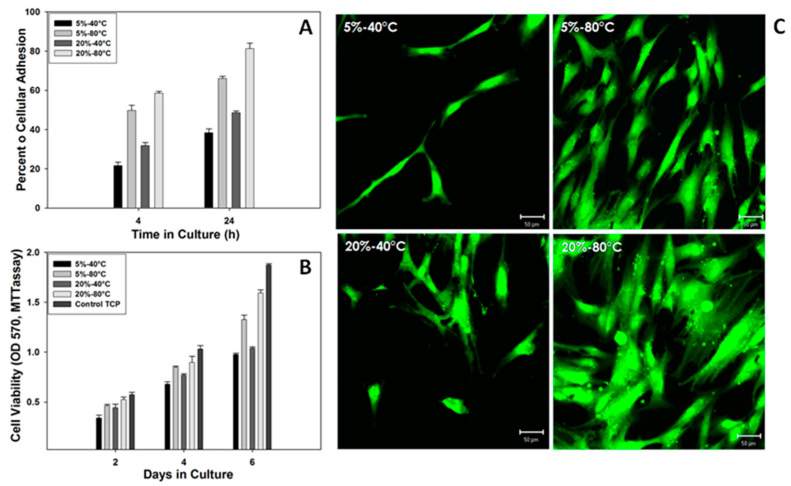
hMSC in vitro culture: (**A**) adhesion, (**B**) viability tests and (**C**) morphology of cell bodies via confocal microscopy.

**Table 1 materials-14-05232-t001:** Root-mean-square height of the surface in terms of intra-/interspherulitic domains.

	Rq Intra-Domain * (nm)		Rq Inter-Domain ** (μm)	
	5%	20%	5%	20%
40 °C	14 ± 3	13 ± 2	0.9 ± 0.1	0.42 ± 0.07
80 °C	65 ± 11	9 ± 2	1.5 ± 0.1	1.6 ± 0.1

* Evaluated from the average on n 1 × 1 μm^2^ ROI boxes. ** Evaluated on 25 *×* 25 μm^2^ ROI boxes.

## Data Availability

Data are contained within the article.

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
