# Peer review of "In Vitro Cell Interactions on PVDF Films: Effects of Surface Morphology and Polar Phase Transition"

_materials, 2021, doi:10.3390/ma14185232_

Round 1

Reviewer 1 Report

Comments in the attached file.

Author Response

attached find enclosed the answers to the Reviewer 1.

Reviewer 2 Report

This work has a quite poor presentation and logic, however, it can be publishable after a major revision as noted

  • The title looks awkward, try a better one.
  • This work introduces three parameters: morphology, phase transition, and effects on in vitro cell interactions. The authors study the PVDF morphology at 40 and 80 oC and use these samples in vitro culture. However, the phase has been identified by using/studying DSC and for 80 oC samples. It is not a correct way to determine the phase of the samples using DSC (as shown in Figure 4). The author should provide XRD for the characterization of samples at 40 and 80 oC with both concentrations (5% and 20%) to make it more logical.
  • Figure 5. Why 5% PVDF (40 oC and 80 oC) has not been provided in this Figure or event supporting information? In the revised version, please also provide β and γ-phases if the author would like to assign something.
  • Scientific explanations have not been done in results and discussion. The authors tend to read the data rather than provide a scientific paper. The author should provide more discussion when presenting the results. Here, I can give an example: why PVDF casting at 80°C with a high concentration of polymer can give a higher brightness in Figure 6C. Please take a look at all Figures and make a suitable discussion.
  • Make more detailed captions for each figure.

Author Response

Find enclosed the answers to the comments of Reviewer 2

Round 2

Reviewer 2 Report

Although XRD is a good method to characterize the phase of a material. However, the current limitation in the institute has prevented that characterization. Besides, the authors have given much effort to improve the manuscript, as such I recommend publishing this revised version.